# Effects of Harmful Cyanobacteria on Drinking Water Source Quality and Ecosystems

**DOI:** 10.3390/toxins15120703

**Published:** 2023-12-16

**Authors:** Marlena Piontek, Wanda Czyżewska, Hanna Mazur-Marzec

**Affiliations:** 1Institute of Environmental Engineering, University of Zielona Góra, Licealna 9, 65-417 Zielona Góra, Poland; 2Water and Sewage Laboratory, Water and Wastewater Treatment Plant in Zielona Góra, 65-120 Zielona Gora, Poland; 3Department of Marine Biology and Ecology, University of Gdańsk, Al. Marszałka Piłsudskiego 46, 81-378 Gdynia, Poland; hanna.mazur-marzec@ug.edu.pl

**Keywords:** cyanobacterial blooms, water quality indicators, *Dugesia tigrina*, microcystins, cyanopeptides

## Abstract

A seasonal plethora of cyanobacteria in the plankton community can have severe implications, not only for water ecosystems but also for the availability of treated water. The catchment of the Obrzyca River (a source of drinking water) is seasonally exposed to harmful cyanobacterial bloom. Previous studies (2008–2012; 2019) revealed that the most polluted water of the Obrzyca River was Uście, close to the outlet of Rudno Lake (at the sampling point). Therefore, the effect on this lake was specifically examined in this study. Sampling was performed from May to September at that site and from July to September 2020 at Rudno Lake. The conducted analysis revealed a massive growth of *Aphanizomenon gracile*, especially in Rudno Lake. The results showed not only the distinct impact of cyanobacterial bloom on phytoplankton biodiversity but also the presence of microcystins and other cyanopeptides in both sampling points. The maximal total concentration of microcystins (dmMC-RR, MC-RR, dmMC-LR, MC-LR, MC-LY, MC-YR) equaled 57.3 μg/L and the presence of cyanopeptides (aeruginosin, anabaenopeptin) was originally determined in Rudno Lake, August 2021. The presence of these toxins was highlighted in our results for the first time. The same samples from the lake were the most toxic in biotoxicological investigations using the planarian *Dugesia tigrina.* The performed bioassays proved that *D. tigrina* is a sensitive bioindicator for cyanotoxins. The physical and chemical indicators of water quality, i.e., color, temperature, total suspended solids, and total nitrogen and phosphorus, showed a significant correlation among each other and towards cyanobacterial abundance and microcystin concentrations.

## 1. Introduction

Water pollution and global warming are increasing threats to public water resources and therefore have become a widespread environmental problem [1,2,3]. The occurrence of harmful cyanobacterial blooms in raw and untreated water bodies affects a primary source of drinking water supply for Drinking Water Treatment Plants (DWTPs). This is a huge problem in the whole world, not exclusively in Poland [4,5,6,7,8,9,10,11]. Globally, water pollution, especially by nutrients, leads to the development of cyanobacterial blooms and degrades the quality of drinking water significantly. Intensive development of phytoplankton including cyanobacteria deteriorates overall water quality (changes in color, odor, flavor, turbidity, and increase in pH). The ongoing economic costs of harmful cyanobacterial blooms can have a massive impact via the effects on water quality, i.e., limited recreation use, fishing, and property values [3]. This issue may very soon lead to major drinking water crises in large cities globally [3]. The quality of raw or untreated water influences the effectiveness of water treatment and the costs associated with drinking water production and distribution. The global monitoring of water bodies and supply systems for cyanobacteria and cyanotoxins is still not common practice. Numerous pivotal control points necessitate testing, with potential locations encompassing water storage reservoirs or rivers [12]. 

Climate change is expected to increase the occurrence of cyanobacterial blooms not just in summer but also in other seasons [13,14], leading to increased continual and escalating ecological impacts. The increase in water pollution can be observed during the cyanobacterial growth phase, which lasts several days or even several weeks after the end of the bloom and in the phase of cell death and lysis. Dead microorganisms further develop a deposit of secondary pollutants, which, under favorable conditions, return contamination to the water column [15]. Therefore, the need for the removal of cyanobacterial cells from raw or sourced water is required. One of the solutions is the application of microstrainer filtration as a preliminary stage in drinking water treatment. The effectiveness of removal of phytoplankton, including cyanobacteria, may be more than 90% [15,16,17]. 

Based on toxicological bioassays and chemical structures, cyanotoxins can be classified into several groups: hepatotoxins including small cyclic peptides (microcystins, nodularins), cytotoxins including alkaloids (cylindrospermopsins), neurotoxins, i.e., alkaloids (anatoxin-a, homoanatoxin-a, saxitoxins), non-proteinogenic amino acids (BMAAs), and phosphate esters (anatoxin-a(S)) [18]. Cyanobacteria are known to produce several other bioactive compounds, for example, oligopeptides such as anabaenopeptins or aeruginosins [19]. The occurrence of cyanotoxins and their secondary metabolites in water may be particularly dangerous, causing illness and death for animals and humans [18,20,21,22]. Therefore, many countries in the world have legal frameworks and guidance for cyanobacteria and cyanotoxin management in drinking water [23]. A provisional limit value of 1 µg/L microcystin LR (the most frequently occurring and most toxic microcystin congeners) in drinking water was suggested by the World Health Organization (WHO) [24].

The Obrzyca River (a right-hand tributary of the Odra River) constitutes a significant source of drinking water for inhabitants of Zielona Góra, a city in western Poland. The catchment of the river is almost two thousand square km, with Sławskie Lake situated at the beginning of the river. The occurrence of cyanobacterial blooms in the catchment area of the river was a signal to start our research. Previous studies have demonstrated fluctuations in both the occurrence and intensity of cyanobacterial blooms in the river, showing that the occurrence and intensity of cyanobacterial blooms in the Obrzyca River vary. The results obtained indicated a higher frequency of excessive cyanobacterial growth in the Obrzyca River, particularly at the Uście sampling area. Consequently, in this work, we aimed to assess the influence of pollution originating from Rudno Lake, located upstream, and estimate the potential cyanotoxic threat to the river, which serves as a source of untreated drinking water.

Rudno Lake with an area of 163 ha, average depth of 4.05 m, and water volume of 659 × 10^5^ m^3^ is a reservoir highly susceptible to pollution degradation. The lack of thermal stratification, epilimnion being in contact with the entire surface of the lake bottom, and low depth all decrease the lake’s natural defense potential against the degrading effects of pollution. Additionally, 59.1% of the lake‘s catchment area is farmland and only 24.9% of the population uses this sewage network. The introduction of an effective water protection system against the pollution of the lake has been suggested [25]. 

The goals of this study were 

-to evaluate the impact of massive point cyanobacterial bloom pollution on phytoplankton biodiversity;-to obtain an indication of the dominant cyanobacterial species causing blooms in Obrzyca River at points, Uście, and Rudno Lake, thereby presenting a potential cyanotoxic threat;-to present relationships between physical, chemical, and microbiological water quality indicators and cyanobacterial biovolume or cyanotoxin concentrations;-to demonstrate, besides cyanotoxins, the presence of other bioactive cyanobacterial compounds, i.e., anabaenopeptins and aeruginosins (real threat discovered during these tests);-to assess the toxicity of cyanobacterial blooms using a bioassay with *Dugesia tigrina.*

## 2. Results

### 2.1. Phytoplankton Community Composition

The highest percentage proportion both in the river and lake was cyanobacteria with a range from 52.7% to 99.7% (Figure 1). A greater phytoplankton biodiversity was observed in the river. The second-largest phytoplankton group (45.2%) determined in samples was diatoms, especially in May. The genera observed were the following: *Nitschia* sp., *Cyclotella* sp., *Asterionella* sp., *Aulacoseira* sp., and *Synedra* sp. The proportion of other phytoplankton groups was insignificant, lower than 5% at the time of study.

The average phytoplankton biovolume equaled 21.3 mm^3^/L in the river and 631 mm^3^/L in the lake. The highest phytoplankton biovolume of 1662 mm^3^/L occurred in the Rudno Lake in August where only cyanobacteria were observed at the time of study. 

### 2.2. Cyanobacteria Community Composition

The maximal amount of cyanobacteria (1662 mm^3^/L) was recorded in Rudno Lake in August 2020, when *Aphanizomenon gracile* was the most numerous species and chlorophyll a (chl-a) concentration equaled 190 μg/L (Table 1). The predominance of *A. gracile* was also observed in September in the tested lake and from July to September in the river. Cyanobacterial blooms (chl-a above 20 μg/L) were observed during the whole sampling in Rudno Lake and from July to August in Obrzyca River. The averages of the chl-a concentration and the biovolume of cyanobacteria were many times higher in the lake than in the river.

Besides *A. gracile*, other species were observed in tested samples: *Limnothrix redekei*, *Dolichospermum flos-aquae*, *D. compacta*, *Planktothrix agardhii*, *Microcystis aeruginosa*, *M. viridis*, *M. flos-aquae*, *A. flos-aquae*, and *Pseudoanabaena limnetica*. The development over time of cyanobacterial genera/species present in the river and the lake is presented in Figure 2. *Aphanizomenon* sp. was the most dominant in August in the lake and in September in the river, *Limnothrix* from June (in the river) to July (in the lake), *Dolichospermum* in July, *Planktothrix* from May to July (in the river) and in September (in the lake), and *Microcystis* in August (in the river) and in July (in the lake) (Figure 2).

### 2.3. Physical–Chemical and Microbiological Water Quality Indicators

The presented studies confirmed relationships between physical–chemical water quality indicators and cyanobacterial biovolume and the total concentration of microcystins (Table 2).

A significant correlation (*p* < 0.05) was observed between chl-a and color, temperature, total suspended solids, and total nitrogen. Cyanobacterial abundance and microcystins were strongly correlated with temperature, total suspended solids, nitrogen (N-tot), and phosphorus (P-tot). Also, a significant correlation was observed between cyanobacterial abundance and color. There was no observed relationship between microbiological indicators and chlorophyll a, cyanobacterial amount, and microcystin concentration. 

### 2.4. Cyanotoxins and Cyanopeptides

In the analyzed samples (Table 3), variants of microcystins were determined: dmMC-RR, MC-RR, dmMC-LR, MC-LR, MC-YR, and trace amounts of MC-LF and MC-LY. The highest total concentration of intracellular microcystins was noted in Rudno Lake (August) and equaled 57.3 μg/L. In the analyzed samples, the averages of microcystin concentration (dmMC-RR, MC-RR, dmMC-LR, MC-YR) were many times greater in the lake.

In our work, other cyanotoxins, i.e., anatoxin-a and cylindrospermopsin, were not detected. The presence of cyanopeptides (aeruginosin, anabaenopeptin), other than microcystins, was recorded in all samples from the lake and one sample collected from the river (Table 3).

### 2.5. Bioassays

Due to possible synergism or antagonism of cyanotoxins and cyanopeptides, biotest results are expressed as % concentration for extracts. The results refer to the total toxicity of the sample. Five samples collected from July to September from the lake and in July and August from the river were toxic to *Dugesia tigrina*. The remaining samples subjected to testing did not exhibit toxicity to the planarian (Table 4). Within 240 h of testing toxic samples, the LC 50 equaled from 5.4 to 60.3% of the concentration of analyzed extracts. The sample collected from the lake in August was the most toxic. The LC 50 amounted to almost 5.4% of the extract and was estimated as 2.88 μg/L total microcystins. The obtained results demonstrate that the sample from Rudno Lake was more toxic than that from the river. 

## 3. Discussion

### 3.1. Phytoplankton Community Composition

The domination of cyanobacteria was observed in the presented results. A high proportion of cyanobacteria over other phytoplankton species is possible due to their unique abilities, for example, binding atmospheric nitrogen and the accumulation and deposition of nitrogen and phosphorus in cells. The advantage of cyanobacteria over other algae is their wider range of thermal tolerance. In addition, the presence of gas vacuoles in the cells makes it possible to regulate the depth of residence, which is particularly important during summer stratification. Greater resistance to zooplankton pressure is an additional advantage [26]. Cyanobacteria may constitute a food source for herbivorous zooplankton; however, this is poor quality and may even constitute a threat to the consumer’s life. The direct negative effect of cyanobacteria on aquatic animal organisms (e.g., rotifers, cladocerans, copepods) may result from a mechanical disruption of grazers in the filtration process or cause a negative influence on the growth rate or reproduction of zooplankton. Cyanobacteria may directly or indirectly affect freshwater plankton via changes in abiotic conditions (deficiency of oxygen, pH) and allelopathic inhibition of the growth of algae, which are a complete food for zooplankton and also planktonic animals pushing to the lower zones of reservoirs [26,27].

### 3.2. Cyanobacteria Community Composition

*A. gracile* was the dominant species in the presented studies from July to September and *L. redekei* in May and June. The most common genera identified in European freshwater lakes and rivers were *Microcystis*, *Planktothrix*, *Aphanizomenon*, and *Phormidium* [18]. *Microcystis* sp. is the most cosmopolitan cyanobacteria genus which occurs massively in temperate regions, i.e., Poland. The most common species in Polish lakes and retention reservoirs are *M. aeruginosa*, *M. flos-aquae*, *M. wesenbergii*, and *M. viridis*, which are the main components of phytoplankton from August to mid-October. Cyanobacteria of the genus *Planktothrix (Oscillatoriales)* are important producers of microcystins in temperate waters, especially in the northern hemisphere, where they can occur all year round. *Aphanizomenon* sp. is the most frequently observed in the summer months in European and Polish lakes, and the most common species is *A. flos-aquae* [28].

*Chroococcales* including the species *M. aeruginosa* were the most common (88%) in 10 French reservoirs. Of all samples analyzed, 62% contained *Oscillatoriales* dominated by *Planktothrix agardhii*, and 47% of the samples were dominated by *A. flos-aquae*. The samples were collected from June to October 2007–2008 [29]. Seventy different English water bodies were sampled between February and December 2016 as part of the cyanobacterial response program. During the monitoring, representatives of the following genera were present: *Microcystis*, *Dolichospermum (Anabaena*), *Oscillatoria*, *Aphanizomenon*, *Aphanothece*, *Pseudoanabaena*, *Planktothrix*, *Snowella*, and *Merismopedia* [30]. In Finnish fresh and coastal Baltic Sea waters, *Dolichospermum* was the most common genus in toxic and non-toxic blooms. In the water bodies, *Microcystis* and *Aphanizomenon* were the next highest, whereas *Oscillatoria* and *Gomphosphaeria* were occasionally found as dominant genera in the blooms [31]. *Microcystis* spp. was the most frequent genus identified in Italian bathing and recreational water bodies [32] and Greek freshwaters [33]. The presence of *Aphanizomenon* was reported in nearly 50% of the 238 examined water bodies in Poland. Therefore, it is one of the most commonly occurring cyanobacteria in Polish freshwaters. *Aphanizomenon* was observed in lakes and rivers in central-western Poland. Generally, in Polish waters, *Aphanizomenon* tends to occur in association with *Dolichospermum* species [34]. In our studies, it was observed that co-dominance of *A. gracile* and *D. flos-aquae* occurred, for example in the sample from Rudno Lake (Figure 2b,f). *A. gracile* was also observed in 35 lakes in central-western Poland in 2010 [35]. A cyanobacterial bloom co-dominated by *A. gracile* with *Dolichospermum* sp. was observed in Vela Lake (Portugal) at the beginning of June 2006. At the end of June, when the *A. gracile* density began to decrease, another *Synechococcus* sp. increased its abundance. In 2012, there was a documented prevalence of diazotrophic cyanobacteria, particularly *Aphanizomenon* spp., during late spring in Vela Lake [36].

### 3.3. Physical–Chemical and Microbiological Water Quality Indicators

The presented studies confirmed relationships between physical–chemical water quality indicators and cyanobacteria biovolume and the total concentration of microcystins. A significant correlation was observed between chl-a and color, temperature, total suspended solids, and total nitrogen. Cyanobacterial abundance and microcystins were strongly correlated with temperature, total suspended solids, nitrogen (N-tot), and phosphorus (P-tot). Cyanobacterial blooms arise from the swift and excessive proliferation along with the buildup of cyanobacterial biomass on the water surface. The growth of these microorganisms is heavily influenced by environmental conditions, particularly nutrient availability, temperature, and light exposure, owing to their photosynthetic activity. Cyanobacterial growth thrives in water temperatures of 25 °C or higher, leading to the observation of blooms predominantly in late summer in temperate water bodies [37].

In previous studies [13], significant correlations between cyanobacterial biovolume and turbidity, total suspended solids, pH, and total nitrogen were recorded. Also, the relationships between the sum of microcystins and water quality markers, i.e., total nitrogen, phosphorus, pH, turbidity, and total suspended solids, were determined. Napiórkowska-Krzebietke et al. [38] observed that filamentous *Limnothrix* was strongly correlated with oxygen concentration but weaker with total nitrogen and phosphorus. The growth of *Aphanizomenon* was stimulated primarily by nutrients and water temperature, whereas the occurrence of *Planktothrix* was correlated with chl-a. The growth of *P. agardhii* increased only up to 0.10 mg/L total phosphorus content and then decreased. The highest responses of cyanobacterial growth to the nitrogen content were noted at 0.8 and 2.2 mg/L for *P. agardhii* and *A. gracile*, respectively, after which the growth intensities were lower. The potentially toxic filamentous *Pseudoanabaena*, *Aphanizomenon*, *Planktothrix*, and *Limnothrix* were constant dominants in the cyanobacteria-dominated phytoplankton almost year-round. Water temperature played a major role in stimulating the growth of filamentous cyanobacteria. However, nitrogen, phosphorus, and even some zooplankters were significantly related to cyanobacteria [38]. In our work, lower concentrations of total phosphorus (<0.6 mg/L) and nitrogen (<5.0 mg/L) promoted the growth of *P. agardhii*. Dominance of *A. gracile* was observed at higher concentrations of nutrients, 1.72 mg/L and 18.9 mg/L, respectively. The studies performed by Figueiredo et al. have indicated that the peak of the excessive growth is coincident with higher temperature, pH, conductivity, and total suspended solids. The *A. gracile* abundance was significantly related to temperature, chlorophyll a, and total suspended solids [36]. According to Kim et al. [39], *Dolichospermum* was strongly correlated with temperature, pH, and P-tot, while the effect of nitrogen was insignificant. In the present study, *D. flos-aquae* was observed as co-dominant with *A. gracile* in the sample from Rudno Lake, where the temperature of water was high (22.3 °C) and pH was the highest at 9.02. In our work, no relationships between chl-a, cyanobacterial biovolume, microcystins, and microbiological water indicators were observed. The influence of cyanobacterial toxins on bacteria is not fully understood because the scientific literature gives several contradictory statements [40]. The lack of correlation between cyanobacterial biovolume and cyanotoxins could indicate the occurrence of toxic and non-toxic cyanobacterial strains [40].

### 3.4. Cyanotoxins and Cyanopeptides

In the analyzed samples, variants of microcystins were found: dmMC-RR, MC-RR, dmMC-LR, MC-LR, MC-YR, and trace amounts MC-LF and MC-LY. The highest total concentration of intracellular microcystins was noted in Rudno Lake (August) and equaled 57.3 μg/L. In the analyzed samples, the predominance of dmMC-RR, with the maximal concentration equaling 29.8 μg/L, was observed. 

Microcystins were the most commonly detected cyanotoxins across European freshwater ecosystems 58% (198 of 341) and have been found in Germany, France, Italy, Spain, Austria, and Poland [18].

In English waterbodies, the highest microcystin concentrations most frequently occurred between August and October 2016. In cyanobacterial blooms dominated by *Aphanizomenon,* a higher concentration of the MC-RR variant was observed from sites associated with lower rainfall [30]. 

MC-RR was the most abundant and the most frequently observed cyanotoxin in Lake Yerevan (Armenia), where *Microcystis, Dolichospermum,* and *Planktothrix* were the key genera identified during the growing season. The maximal microcystin concentration was observed in August and equaled below 1 μg/L [41]. 

The bloom in Lubosińskie Lake (central-western Poland) was dominated by potentially toxic species: *P. agardhii*, *L. redekei*, and *A. gracile*. The toxin analysis revealed the presence of demethylated forms of microcystin-RR and microcystin-LR at concentrations in the ranges of 24.6–28.7 and 6.6–6.7 μg/L [14].

The Siemianówka Dam Reservoir (north-eastern Poland) is a polymictic reservoir constructed in the upper sector of the Narew River, where toxic cyanobacterial blooms occur. The dominance of *P. agardhii* and the presence of *A. gracile* were observed (2010–2011). Microcystin concentrations were in a range of 1.68–16.2 μg/L and mainly MC-RR and dmMC-RR were detected in four depth layers (0.5; 2.0; 4.0; 6.0 m) [42].

In our work, other cyanotoxins, i.e., anatoxin-a and cylindrospermopsin, were not detected. The presence of cyanopeptides (aeruginosin, anabaenopeptin) was recorded in all samples from the lake and in only one sample collected from the river in August. 

Cyanobacteria, like other groups of bacteria, produce a wide range of non-ribosomal peptides, including anabaenopeptins and aeruginosins [19]. These metabolites were identified in bloom samples from fresh and marine waters. Anabaenopeptins are active toward proteolytic enzymes such as chymotrypsin, elastase, and carboxypeptidase-A. Due to this kind of activity, the peptides may play some role in the interaction with other aquatic organisms, e.g., as elements of chemical defense against grazers or infecting agents [19]. Recently, neurotoxic (inhibition of catecholamine neurotransmitter activity) and cytotoxic (degradation of F-actin) effects were observed with anabaenopeptins from *Brachionus calyciflorus* [43]. 

The production of microcystins, aeruginosins, and anabaenopeptins or even specific variants of the peptides is common among European *P. agardhii* populations, including those from Polish water bodies. In the bloom of the cyanobacteria, the frequency of aeruginosins appeared to be lower than microcystins and anabaenopeptins [42].

### 3.5. Bioassays

A majority (62.5%) of tested samples were toxic to *Dugesia tigrina*. Amid toxic samples tested for 240 h, LC 50 was in the range of 5.4–60.3% of the concentration of the analyzed extracts. The samples collected from the lake in August were the most toxic. The LC 50 amounted to almost 5.4% of the extract and was estimated as 2.88 μg/L total microcystins. The obtained results demonstrate that the sample from Rudno Lake was more toxic than that from the river.

In our previous analysis (river samples collected in 2019), the most toxic (20.9%) was the bloom sample collected in September when the sum of microcystins was more than 3 μg/L [13]. The planarian *Dugesia tigrina* proved to be a sensitive cyanotoxin bioindicator. Planarians are more sensitive than the daphnids often used for similar purposes [13].

The toxic influence of cyanobacterial blooms with microcystins for the survival of *Daphnia pulex* was analyzed by Nandini et al. The LC 50 was 26.8 μg/L and 11.5, respectively, for *M. aeruginosa* and *Woronichinia naegeliana* specimens [44]. 

Results of acute and chronic toxicity tests of cyanobacterial extracts with *Daphnia magna* were also described by Freitas et al. The 48 h LC 50 for hepatotoxins ranged approx. from 217 to 270 mg/L and from 211 to 235 mg/L for neurotoxins. The 24 h LC 50 for the feeding inhibition was slightly higher for the hepatotoxic extracts (approx. 117–145 mg/L) than neurotoxic (approx. 79.5–107 mg/L) [45]. 

Results of acute toxicity tests with *Daphnia similis* were presented by Herrera et al. Daphnids were exposed to natural cyanobacterial bloom samples and results have shown more than 50% of individuals ceased to survive after 24 h of exposure at a concentration of 250 mg DW/L (dry weight per liter). Toxicity significantly increased since LC 50 changed from 486 to 175 mg DW/L after 24 h exposure [46].

The acute toxicity tests performed by Wan et al. showed that the 48 h LC 50 for MC-LR to *Daphnia magna* was 20.3 mg/L [47]. The results with daphnids mentioned above are confirmation that these organisms are less sensitive than planarians. 

Acute toxic effects of microcystins and saxitoxins were detected using the cladocerans *Moina micrura* by Ferrão-Filho et al. Samples collected from the Funil reservoir (Rio de Janeiro) from November to December were the most toxic, and the survival rate decreased with the concentration of raw water. Values of 24 h LC 50 equaled 91.2% (1.09 µg/L equivalent MC-LR) in a sample collected in November and 79.4% (3.57 µg/L equivalent MC-LR) in December [48]. 

The bioassays of samples (aquatic extracts of cyanobacterial blooms predominated by *P. agardhii)* with the use of young duckweed *Spirodela polyrhiza* revealed a negative effect on its biomass and root system. The examined extracts differed considerably in the content of MCs and other oligopeptides, i.e., anabaenopeptins and aeruginosins [49]. Also, cyanobacterial anabaenopeptin-B (AN-B) inhibits *Daphnia magna* swimming behavior similarly to the microcystins MC-LR and MC-LF. The mixture of anabaenopeptin-B and MC-LR caused stronger toxic effects than the individual oligopeptides used at the same concentration. The much lower 48 h EC 50 value of the AN-B and MC-LR mixture (0.95 ± 0.12 μg/mL) than those of the individual oligopeptides AN-B (6.3 ± 0.63 μg/mL), MC-LR (4.0 ± 0.27 μg/mL), and MC-LF (3.9 ± 0.20 μg/mL) that caused swimming speed inhibition explains the commonly observed stronger toxicity of complex crude cyanobacterial extracts to daphnids than individual microcystins [50].

## 4. Conclusions

The conducted analysis allows the following conclusions to be drawn:Hydrobiological and physicochemical analysis revealed:
the Rudno Lake is highly eutrophic and contains the most dangerous cyanotoxic point of pollution in the Obrzyca River, which is a primary source of drinking water for the inhabitants of Zielona Góra;in the collected samples with cyanobacterial blooms, a predominance of *Aphanizomenon gracile* was observed to be correlated with higher water temperature;the highest cyanobacterial biovolume and chl-a concentration were noted in August samples in Rudno Lake and equaled 1662 mm^3^/L and 190 μg/L, respectively;significant correlations were observed between chl-a concentration and color, temperature, total suspended solids, and total nitrogen;the water suitability indicators, i.e., total nitrogen, phosphorus, total suspended solids, and temperature, were strongly related to cyanobacterial biovolume and total microcystin concentration;statistical analysis revealed the impact of cyanobacterial pollution in Rudno Lake on the quality of the Obrzyca River.
Chromatographic analysis has shown that:
in analyzed samples, cyanopeptides (aeruginosins and anabaenopeptins) were detected for the first time;the highest total microcystin concentration was noted in the sample with the maximal chl-a concentration (190 μg/L) and equaled 57.3 μg/L.
Bioassays have shown:
The most toxic sample for *D. tigrina* was extracted from the lake where the maximal cyanotoxin concentration was determined. LC 50 equaled about 3 µg/L MCs;the planarian proved to be a sensitive cyanotoxin bioindicator and can be used in the toxicological analysis of cyanobacterial blooms.

The Rudno Lake is highly eutrophicated and it is the point of pollution of the Obrzyca River. Lake water management has still not been implemented and the problem is still ongoing. Many years of neglect in the water and sewage management in the catchment area of Lake Rudno led to its permanent degradation; symptoms include blooms of cyanobacteria, accumulation of large deposits of bottom sediments, impoverishment of the lake’s biodiversity, decrease in transparency, and, consequently, decline in recreational attractiveness. In recent years, the poor ecological status of the lake waters, influenced by phytoplankton, has been persistent, so the removal of harmful cyanobacteria should become an important objective in water management strategies implementing the following methods: (1) biological—especially with floating treatment wetlands and riparian vegetation; (2) physical—with aeration, mechanical circulation, and hypolimnetic oxygenation; (3) chemical—coagulation and flocculation processes; (4) mechanical methods—micro-sieving process; the last method has been applied and demonstrated in previous research [51]. 

## 5. Materials and Methods

### 5.1. Study Area and Sampling Sites

A total of eight samples were taken from the Obrzyca River in the west of Poland at Uście point (52°00′39″ N 15°56′48″ E) from May to September 2020 and from the Rudno Lake (52°00′13″ N 15°57′44″ E) from July to September 2020. The sample points were chosen based on the results of previous cyanotoxic threat analyses in the Obrzyca River [9,13]. The points and distance from DWTP are marked in Figure 3. Samples for microscopic and toxicological analyses were filtrated using a mesh plankton diameter of 10 µm and stored in 0.25 L bottles. One-liter collected samples without filtration were intended for chromatographic analysis [9,13]. 

### 5.2. Phytoplankton Qualitative and Quantitative Methods

Hydrobiological analyses were performed in triplicate, utilizing a Sedgewick–Rafter chamber and an MN 358/A microscope (OPTA TECH, Warszawa, Poland). Biovolume results were expressed as mm^3^/L [13]. 

### 5.3. Physical and Chemical Water Quality Indicators

The water quality physical sample (color, turbidity, temperature) and chemical indicators (ammonium nitrogen (NH_4_), total nitrogen (N-tot), orthophosphate (PO_4_), total phosphorus (P-tot), total suspended solids (TSS), dissolved oxygen (DO)) were determined in DWTP. Measurement of the markers was performed according to ISO and HACH Standards [52,53,54,55,56,57,58,59,60]. 

### 5.4. Microbiological and Biological Water Quality Indicators

*Escherichia coli*, *Clostridium perfringens*, and *Enterococcus faecalis* were chosen as microbiological water quality indicators. The measurement of these bacteria was performed according to ISO Standards [61,62,63] using a membrane filtration technique.

The measurement of chlorophyll a was performed using the spectrophotometric method. 500-milliliter samples were filtered through a 47 mm fiberglass filter (GF/C) followed by extraction with 90% acetone for 24 h and subsequent centrifugation for 10 min. at 4000 rpm. The spectrophotometric measures were carried out using a spectrophotometer DR 2000 (HACH, Loveland, CO, USA). The absorbance measurements and calculations of chlorophyll a concentration were conducted according to PN-C-05560-02 [64].

### 5.5. Cyanotoxin Analysis

Water samples (250–500 mL) for cyanotoxin analysis were filtered through Whatman GF/C filter discs and stored at −20 °C. The material gathered on the filter was extracted using the efficient and recommended solvent −75% methanol in water by probe sonication (1 min, 20 kHz, 25% duty cycle) [65,66]. After centrifugation (10,000× *g*; 15 min), the supernatants were separated and analyzed using a chromatographic system (Agilent Technologies, Boeblingen, Germany) equipped with a tandem mass spectrometer (QTRAP5500, Applied Biosystems, Sciex, Concorde, ON, Canada). The separation was performed with a Zorbax Exlipse XDB-C-18 column (4.6 × 150 mm, 5 µm; Agilent Technologies, Santa Clara, CA, USA) using gradient elution (0.6 mL/min) with 5% acetonitrile (phase A) and 100% acetonitrile (phase B), both with 0.1% formic acid. For quantitative analyses, microcystins (MCs), anatoxin-a (ANA), cylindrospermopsin (CYN), and nodularin (NOD) standards (Alexis Biochemicals, San Diego, CA, USA) were used.

### 5.6. Bioassay Tests

The gathered specimens were measured volumetrically and subsequently centrifuged at 4000 rpm for 20 min. The resulting precipitates underwent to three freeze-and-thaw cycles to lyse the intact cells and release the intracellular toxins [9]. Microscopic examination was used to confirm cell lysis. The toxicological tests were conducted for 8 collected specimens using the planarian *Dugesia tigrina*. Lethal concentration (LC 50) was calculated and results were presented as % concentration of extracts. The planarian cultivation method used for toxicological experiments was published by Piontek. The use of the planarian in the toxicological investigations was also described by Piontek and a detailed explanation was given in Czyżewska et al. [13].

### 5.7. Statistical Analysis

The Excel 2010 program was utilized to perform Pearson’s correlation analysis with simultaneous comparison at the significance level α = 0.05. Statistical analysis of the experimental results on the toxic effect of extracts was carried out by the probabilistic graphical method Probabilistic Graphical Models [67]. The method was used to calculate the value of LC 50. The results obtained underwent a test for the conformity of the experimental distribution with the normal distribution, using the χ^2^ test for calculation. The tested distributions were deemed sufficiently convergent with the normal distribution if the probability in the χ^2^ test exceeded 0.7. A detailed description was gbyen in Czyżewska et al. [13]. 

## Figures and Tables

**Figure 1 toxins-15-00703-f001:**
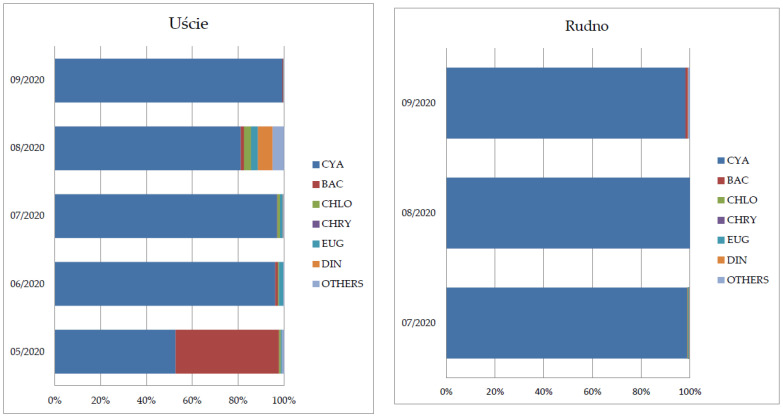
Shares of single groups of plankton in the Obrzyca River (Uście) and Rudno Lake (CYA—cyanobacteria, BAC—diatoms, *Baccilariophyceae*, CHLO—*Chlorophyta*, CHRY—*Chrysophyceae*, EUG—*Euglenoidea*, DIN—*Dinophyceae*, OTHERS—zooplankton: ciliates, rotifers, copepods).

**Figure 2 toxins-15-00703-f002:**
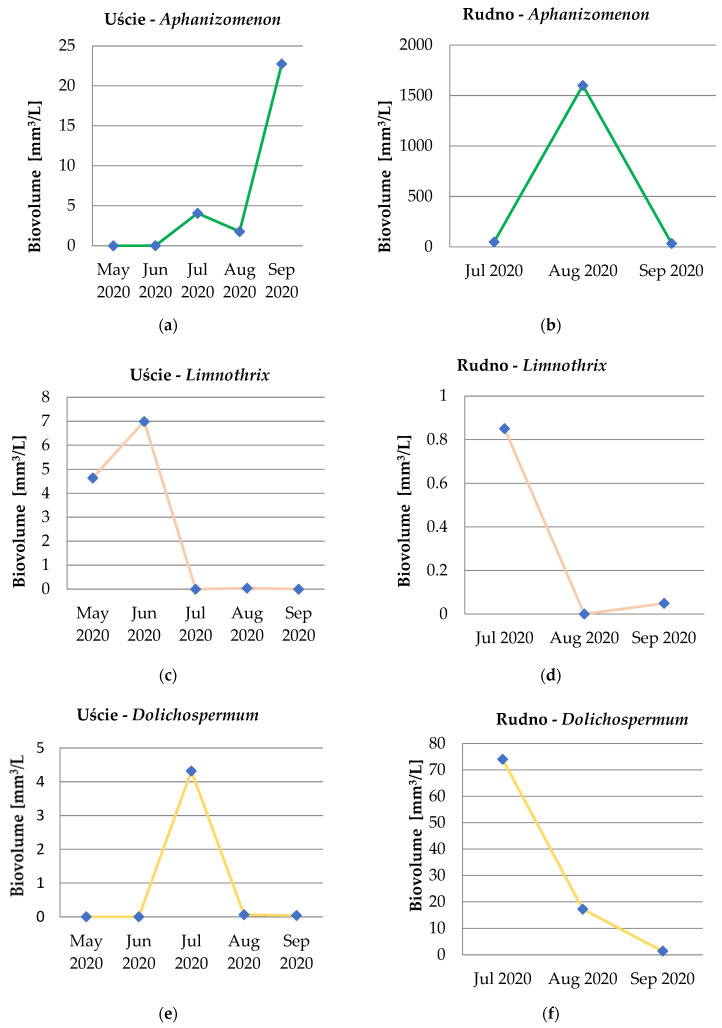
Dynamics of dominating cyanobacteria genera in the Obrzyca River (**a**,**c**,**e**,**g**,**i**) and Rudno Lake (**b**,**d**,**f**,**h**,**j**) (the charts are presented in different scales).

**Figure 3 toxins-15-00703-f003:**
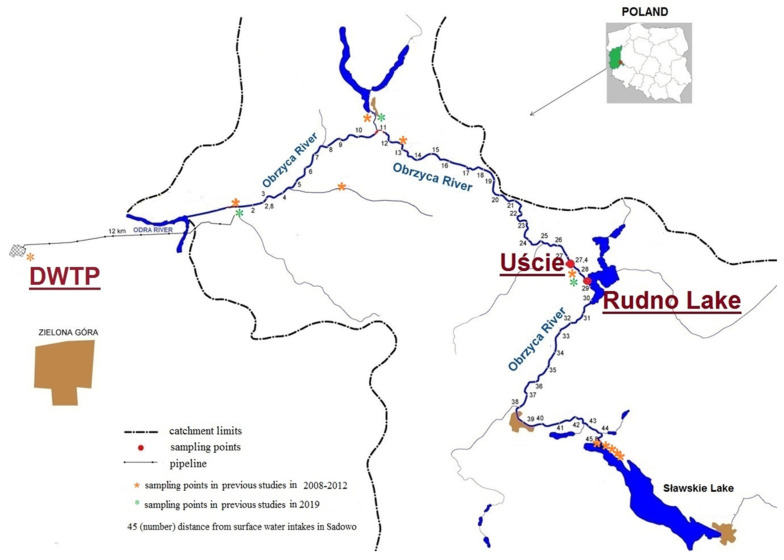
Sampling points.

**Table 1 toxins-15-00703-t001:** Biovolume of cyanobacteria with dominant species and chl-a concentration in Obrzyca River at Uście point and Rudno Lake.

Sampling Month	UŚCIE			RUDNO		
	Chl-a [μg/L]	Biovolume [mm^3^/L]	Dominant Species	Chl-a [μg/L]	Biovolume [mm^3^/L]	Dominant Species
05/2020	23.5	4.81	*L. redekei*	-	-	-
06/2020	46.7	7.23	*L. redekei*	-	-	-
07/2020	69.9	26.1	*A. gracile*	129	177	*D. flos-aquae* *A. gracile*
08/2020	21.4	22.9	*A. gracile*	190	1662	*A. gracile*
09/2020	9.60	5.32	*A. gracile*	37.4	37.2	*A. gracile*
Average	34.2	13.3	*-*	118.8	625.4	*-*

**Table 2 toxins-15-00703-t002:** Values of Pearson correlation coefficients between water quality indicators (physical–chemical and microbiological) and cyanobacterial abundance.

Water Quality Indicator	Chl-a	CYA	∑MCs
pH	0.69	0.56	0.53
Color	0.72 *	0.73 *	0.57
Temperature	0.71 *	0.74 *	0.72 *
Turbidity	0.11	−0.29	−0.25
Total suspended solids	0.85 *	0.99 *	0.99 *
Dissolved oxygen	0.47	0.09	0.06
N-tot	0.71 *	0.91 *	0.89 *
NH4	−0.58	−0.50	−0.43
P-tot	0.35	0.71 *	0.71 *
PO4	−0.15	0.15	0.18
**Microbiological indicators**			
*Escherichia coli*	−0.27	−0.27	−0.25
*Enteroccus faecalis*	−0.42	−0.30	−0.15
*Clostridium perfringens*	−0.17	−0.03	−0.28

*—statistically significant correlation coefficients (*p* < 0.05), CYA—cyanobacterial biovolume.

**Table 3 toxins-15-00703-t003:** Cyanotoxin and cyanopeptide concentration in the Obrzyca River and Rudno Lake.

Sampling Months	CYANOTOXINS [μg/L]												CYANOPEPTIDES [pg/L]	
	ANT	CYN	dm MCRR	MCRR	MCLA	dm MCLR	MCLF	MCLR	MCLY	MCLW	MCYR	∑MCs	ANAB	AER
**UŚCIE**														
05/2020	n.d.	n.d.	0.53	0.01	n.d.	0.16	n.d.	0.02	n.d.	n.d.	0.08	**0.80**	n.d.	n.d.
06/2020	n.d.	n.d.	0.32	0.01	n.d.	0.26	n.d.	0.02	n.d.	n.d.	0.19	**0.80**	n.d.	n.d.
07/2020	n.d.	n.d.	1.11	0.02	n.d.	0.92	n.d.	n.d.	n.d.	n.d.	0.64	**2.75**	n.d.	n.d.
08/2020	n.d.	n.d.	0.003	0.003	n.d.	n.d.	n.d.	0.02	n.d.	n.d.	0.003	**0.21**	11.2	0.60
09/2020	n.d.	n.d.	n.d.	0.003	n.d.	n.d.	n.d.	n.d.	n.d.	n.d.	n.d.	**<0.01**	n.d.	n.d.
**Average**	-	-	0.39	0.04	-	0.27	-	0.01	-	-	0.18	**0.91**	2.24	0.12
**RUDNO**														
07/2020	n.d.	n.d.	1.53	0.02	n.d.	1.43	n.d.	n.d.	n.d.	n.d.	1.22	**4.23**	n.d.	0.60
08/2020	n.d.	n.d.	29.8	0.28	n.d.	13.2	0.02	n.d.	0.02	n.d.	12.1	**57.3**	1.53	9.05
09/2020	n.d.	n.d.	1.68	0.03	n.d.	1.02	n.d.	n.d.	n.d.	n.d.	0.54	**3.34**	9.0	6.82
**Average**	-	-	11.0	0.11	-	5.23	-	-	-	-	4.62	**21.6**	3.51	5.49

n.d.—not detected, ANT—anatoxin, CYN—cylindrospermopsin, ANAB—anabaenopeptins, AER-aeruginosins.

**Table 4 toxins-15-00703-t004:** LC 50 of the biotoxicological test using planarian *Dugesia tigrina*.

Sampling Months	UŚCIE	RUDNO
05/2020	n.t.	n.a.
06/2020	n.t.	n.a.
07/2020	60.3 *	15.5 *
08/2020	42.7 *	5.4 *
09/2020	n.t.	21.4 *

n.t.—not toxic, n.a.—not analyzed, * % concentration of cyanobacterial extracts.

## Data Availability

Data availability is possible after contact with authors.

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
