# Peer review of "Effects of Harmful Cyanobacteria on Drinking Water Source Quality and Ecosystems"

_toxins, 2023, doi:10.3390/toxins15120703_

Round 1

Reviewer 1 Report

Comments and Suggestions for Authors

Conscientiously elaborated study. I especially appreciate that authors paid attention to tiny filamentous  planctic species of cyanobacteria (Limnothrix, Pseudanabaena ane also Aphanizomenon gracile) that were in toxicology overlooked (compared with huge water bloom forming colonies of Microcystis, Dolichospermum and Planktothrix). Please,  change  the name "Melosira" to "Aulacoseira" (line 106) according to modern taxonomy (planktic species e.g. A. granulata, A. ambigua etc.). 

Author Response

Dear Reviewer,

thank you for your valuable comments.

The  change of Aulacoseira name was made.

with kind regards

Reviewer 2 Report

Comments and Suggestions for Authors

Authors have presented research aimed at evaluating the impact of water pollution and cyanobacterial blooms on public water sources, specifically focusing on the Obrzyca River and Rudno Lake in Poland. The study aims to evaluate the impact of pollution on phytoplankton biodiversity, identify dominant cyanobacteria species causing blooms, examine relationships between water quality indicators and cyanobacteria biovolume or cyanotoxins concentrations, assess the presence of other cyanobacterial bioactive compounds, and evaluate the toxicity of cyanobacterial blooms using bioassays.

The methods of analysis appear appropriate, and the results do not appear to be over-interpreted. However, some information is unclear and it should be widely checked. I would suggest a minor revision of the manuscript prior publication.

SPECIFIC COMMENTS FOR REVISION

  • I have identified several typographical errors in the document, including issues with spaces, missing commas, dots, and inconsistent use of italics. Additionally, there are variations in font styles and sizes throughout the manuscript. I kindly request that you thoroughly review and correct these errors.
  • The introduction section would greatly benefit from providing a clear and concise overview of the topic. To achieve this, I suggest considering the rephrasing of sentences to make them more concise and focused. Furthermore, reorganizing the information in a manner that creates a logical flow will make it easier for the reader to follow.
  • The quality of the figures needs improvement. For instance, in Figure 2, the letters appear too small, while the figures themselves seem too large. Additionally, none of the graphs have consistent formatting. It is also worth noting that there are typographical errors present, such as the absence of square brackets for Figure 2G. I recommend addressing these issues accordingly.
  • I kindly request the removal of the yellow markers in Table 2.
  • Regarding Table 4, please provide an explanation of how the data is expressed.
  • In line 173, the authors refer to "Table 5," which is not included in the manuscript. Please rectify this discrepancy.
  • The conclusion section appears excessively long, consisting of nine bullet points and a lengthy paragraph. I recommend rewriting this section to make it more concise and to the point.
  • The resolution of Figure 3 requires significant improvement, as it currently appears blurry and difficult to read. Please take measures to enhance its clarity.
  • The materials and methods section lacks sufficient information in certain areas. Specifically, in section 5.6 regarding the bioassay test, there is mention of "toxicological tests using Dugesia tigrine." I kindly request a proper explanation of this procedure. Additionally, the font and size used in this section differ from the rest of the manuscript, which should be rectified for consistency.
  • In section 5.7, it would be beneficial to understand why the authors chose to perform a Pearson's analysis and Chi-squared tests. Furthermore, I recommend utilizing specific statistical software to conduct these studies for enhanced accuracy and reliability.

Author Response

Dear Reviewer,

thank you for your valuable comments.

According to your comments I made improvements written below:

  • reviewed and corrected  typographical errors in the document, including issues with spaces, missing commas, dots, and inconsistent use of italics.
  •  standardized font styles and sizes throughout the manuscript.
  • introduction was rephrased
  • removed yellow markers in Table 2
  • Figure 2 was improved
  • added explanation of data expression in Table 4
  • corrected a numeration of table 4 in the text
  • conclusions were grouped
  • improved resolution of Figure 3
  • added information in section 5.7 concerning use of Chi-squared test
  • Regarding Table 4, please provide an explanation of how the data is expressed. - the explanation was added at the beginning point 2.5 of the manuscript

Reviewer 3 Report

Comments and Suggestions for Authors

The authors present a method concerning the analysis of a range of cyanotoxins in a drinking water source. The main concern is that only 70% methanol was used to extract toxins for analysis. This may have profound differences on the extractability of some cyanotoxins and may influence the results. In addition the following should be addressed before publication:

1. Aph. gracile- throughout the manuscript change to A. gracile.

2. Figure 2- can these graphs be combined to save on space?

3. Table 1- italicize cyanobacterial names in legend.

4. Table 2 and throughout the manuscript- change the number so that "," is replaced by "."

5. Table 4- what do the numbers mean? Are these dry weights extracted? Please elaborate.

6. Check "i.a." and change to "i.e."

7. Make sure "et al." has "." and "," where required.

8. Line 388: Did you do any comparisons of the effect of pure microcystins, extracts and mass spec analysis of the extracts? 

Comments on the Quality of English Language

Although this reviewer appreciates that English is not the authors first language, some English editing would improve the manuscript.

Author Response

Dear Reviewer,

thank you for your valuable comments.

According to your comments I made improvements written below:

  • Aph.gracile  was changed to A. gracile
  • Figure 2 - it`s impossible to combine the charts because of various scales 
  • Table 1 - it was italicized Latin names 
  • corrected typographical errors, changed "i.a" to "i.e", et al with point at the end
  • Table 4- what do the numbers mean? Are these dry weights extracted? Please elaborate -  the explanation was added at the beginning of point 2.5 of the manuscript
  • the manuscript was revisioned by English native speaker.
  • " Did you do any comparisons of the effect of pure microcystins, extracts and mass spec analysis of the extracts? " We didn`t. Analysis of  pure toxins would be testing the sensitivity of the bioindicator to selected toxins, not assessing the total toxicity of the samples we tested.
  • "The main concern is that only 70% methanol was used to extract toxins for analysis. This may have profound differences on the extractability of some cyanotoxins and may influence the results." It was applicated 75% methanol in this study. Application of the solvent was recommended in many publications e.g.: Handbook of Cyanobacterial Monitoring and Cyanotoxin Analysis, First Edition.
    Edited by Jussi Meriluoto, Lisa Spoof and Geoffrey A. Codd.
    © 2017 John Wiley & Sons, Ltd. Published 2017 by John Wiley & Sons, Ltd.
  •